# Valorisation of Spent Yeast Fermentation Media through Compositional-Analysis-Directed Supplementation

Laura Murphy [1,2,†], Ciara D. Lynch [1,2,†] and David J. O'Connell [1,2,*]

1   School of Biomedical and Biomolecular Sciences, University College Dublin, Belfield, D04 V1W8 Dublin, Ireland; laura.murphy1@ucdconnect.ie (L.M.); ciara.lynch.4@ucdconnect.ie (C.D.L.)
2   BiOrbic, Bioeconomy Research Centre, University College Dublin, Belfield, D04 V1W8 Dublin, Ireland
*   Correspondence: david.oconnell@ucd.ie
†   These authors contributed equally to this work.

**Abstract:** Spent fermentation media from bioprocessing represent a significant waste stream, and interest in recycling them as part of the developing circular bioeconomy is growing. The potential to reuse yeast spent culture media (YSM) to feed secondary bacterial fermentations producing recombinant protein was investigated in this study. Elemental and amino acid compositional analysis using inductively coupled plasma mass spectrometry (ICP-MS) and LC-MS/MS identified significant differences in the concentrations of 6 elements and 18/20 amino acids in YSM compared with rich microbiological media (LB). Restoration of levels of magnesium and sodium through addition of their salts and amino acids from tryptone supplementation led to the expression of equivalent titres of recombinant proteins by *E. coli* (0.275 g/L), compared to that in LB media (0.296 g/L) and BMMY media (0.294 g/L) in shake flask culture. When this supplementation strategy was employed in a bioreactor system, we observed a significant increase in recombinant protein titre using the supplemented YSM (2.29 ($\pm$0.02) g/L) over that produced using LB media (1.29 ($\pm$0.09) g/L). This study demonstrates through highly sensitive compositional analysis and identification of supplementation strategies the potential to valorise spent media from yeast fermentations that underpin industrial processes of significant scale, creating a circular approach to waste stream management.

**Keywords:** *Komagataella phaffii*; *Pichia Pastoris*; waste media; circular bioeconomy; *Escherichia coli*; valorisation; elemental analysis; fermentation; recombinant protein production; supplementation; ICP-MS

## 1. Introduction

The implementation of a circular bioeconomy is at the forefront of developing sustainable practices across industrial processes. The fundamental aims are to incorporate a 'loop' in which materials and by-products are reused or repurposed into other processes [1,2]. While a prominent goal in sustainability practices is to reduce the volume of waste that is produced by industries and processes, dealing with the large amounts of waste currently produced by industrial processes is where the circular bioeconomy can have an immediate impact. One process-intensive area producing significant volumes of waste not currently reused is in the production of biopharmaceuticals, which generates significant volumes of cell culture media waste [3]. A study into the life cycle of these media found that their cost dominated the operating cost of monoclonal antibody production, particularly with regard to continuous culturing methods such as perfusion due to the higher consumption of media in this bioprocess system [4]. Developing strategies to reuse this waste stream could both reduce the operational cost and counteract the environmental impact of biopharmaceutical production.

The main component of media is water for injection (WFI), which is energy-intensive to produce as it requires treatment such as distillation at high temperatures. It has been found that to produce 1 kg of antibody for example, it takes 7700 kg of input on average

from raw materials, and over 90% of this is from water alone [5]. This number came from a study into the process mass intensity (PMI) of the biopharmaceutical industry with six participating companies for benchmarking. This PMI is mainly used as a measurement of the resource efficiency of a process, and not of the environmental impact [6]. Another defining metric for water usage in the biotech industry is that of WAter Related Impact of ENergy (WARIEN). This calculation was used in 2020 to find the $CO_2$ emitted per kilogram of biopharmaceutical product, including all water cleaning methods such as distillation [7]. They identified the WARIEN as ranging from 16 to 80 kg per kg antibody. The WARIEN values for continuous production of recombinant proteins in industry could be substantially reduced by reusing waste media, as media have been shown to be the largest contributor to the high WARIEN values associated with this type of bioprocess, with around 95% of the PMI for water being media-related [7]. The feasibility of this circular model has already been demonstrated with both mammalian and fungal spent cell media, both of which achieved equivalent levels of productivity in a secondary bacterial culture as the standard rich medium, lysogeny broth (LB) [8,9].

Yeast-based bioprocesses have been in common usage for thousands of years with processes of alcohol and food production with baker's yeast (*Saccharomyces cerevisiae*) [10,11]. In the last fifty years since the advent of genetic engineering, yeast has also been identified as a useful eukaryotic cell culture for building recombinant proteins for the pharmaceutical industry such as human insulin for the treatment of diabetes, the majority of which is produced in S. cerevisiae [12–14]. The first recombinant human antibody to be produced in the yeast species *Komagataella phaffii* (also known as *Pichia pastoris*), namely Eptinezumab, an antibody with anti-migraine effects, was given FDA approval in 2020 [15]. *K. phaffii* has now become a common platform for recombinant protein expression, with five biopharmaceuticals produced by this organism approved for human use in the last five years alone [16]. Of the 159 recombinant biopharmaceuticals that were approved in 2018–2022, 9 were made with yeast-based systems, compared to the 4 recombinant biopharmaceuticals made with yeast out of the 62 total approved in the previous four years [16,17].

A standard medium for yeast cultivation is BMMY (buffered methanol-complex medium) [18,19]. This medium is principally used for the inducible expression of biomolecules using methanol as the inducer while cultivating the native methanotroph *K. phaffii*. Here, we use a primary *K. phaffii* culture to generate spent media for analysis. The spent BMMY media (YSM) are then used as fermentation feed in a secondary bacterial culture. Metabolic and elemental composition analysis of the spent media was performed to identify changes in levels of amino acids and elements compared with a standard microbiological medium (LB) to inform the required supplementation steps to efficiently valorise the YSM.

## 2. Materials and Methods

### 2.1. Generation of Spent BMMY for Use in Secondary Fermentations

*Komagataella phaffii* is a common biotechnology yeast species used for recombinant protein production [20]. *K. phaffii* strain GS115 was grown on yeast extract peptone dextrose (YPD) agar (1% yeast extract, 2% peptone, 2% glucose and 15% agar). YPD medium (50 mL) was inoculated with a single GS115 colony and grown for 18 h on an orbital shaker at 250 RPM at 30 °C until an $OD_{600\,nm}$ of between 2 and 6 was reached. Cells were harvested by centrifugation at $3000\times g$ for 5 min at room temperature. The pellet was resuspended in buffered methanol-complex medium (BMMY) (2% peptone, 100 mM potassium phosphate pH 6.0, 1.34% Yeast Nitrogen Base, $1 \times 10^{-5}$% biotin, 0.5% methanol) to an $OD_{600\,nm}$ of 1.0 (approximately 400 mL) in a baffled 2 L Erlenmeyer flask [21]. The culture was grown at 30 °C with shaking at 250 RPM for 48 h, with methanol added to 0.5% *v/v* at T24 to maintain levels of methanol. The final culture medium was harvested via centrifugation at $3000\times g$ for 5 min. Supernatant samples were taken for inductively coupled plasma mass spectrometry (ICP-MS) analysis and metabolomic analysis (LC-MS/MS) pre- and post-fermentation. The spent BMMY media were frozen at −20 °C prior to use for supplementation studies.

## 2.2. Inductively Coupled Plasma Mass Spectrometry (ICP-MS) Analysis of Media Samples

First, 1 mL of clarified media (fresh LB, fresh BMMY, spent BMMY post-yeast culture and spent BMMY post-bacterial culture) was diluted in a 10 mL solution with a final nitric acid concentration of 2% [22]. This was sterile filtered (0.22 μm filter, Millipore, Cork, Ireland) and labelled. Samples were analysed using a quadrupole instrument (Thermo-Fisher Scientific, Waltham, MA, USA) equipped with CCTED (collision cell technology with kinetic energy discrimination) at the University of Nottingham. External calibration standards were used to prime the instrument prior to analytical sample loading. Sample processing was undertaken using 'Qtegra software' (Thermo-Fisher Scientific). Results are reported as gravimetric concentrations (μg/Lor mg/L) in Microsoft Excel spreadsheet format.

## 2.3. Metabolomic Analysis of Amino Acid Content via LC-MS/MS

The individual media samples were sterile filtered with a 0.22-micron filter and immediately frozen at $-80\ °C$ and submitted for analysis. The instrument used for metabolomic analysis was a SCIEX QTRAP 6500 plus mass spectrometer, (SCIEX, Framingham, MA, USA) coupled to SCIEX ExionLC™ Series UHPLC capability (Conway Institute UCD core facility).

## 2.4. Preparation and Supplementation of Spent BMMY for Secondary E. coli Fermentations

Spent BMMY media were thawed at room temperature, sterile filtered (0.2 μm filter, Nalgene™ Rapid Flow™, SFCA membrane, Thermo-Fisher Scientific, Waltham, MA, USA) and adjusted to pH 7 with sodium hydroxide. As determined by ICP-MS and LC-MS/MS data, either 2 mM magnesium sulphate, 14 mM NaCl or 1% tryptone (*w/v*) was added to the spent media as supplements alone or in combination.

## 2.5. Expression and Purification of mCherry-EF2

For all conditions, following transformation, a single mCherry-EF2 colony [23,24] was chosen and inoculated in a 100 mL mixture of lysogeny broth (LB), 2% glucose and ampicillin at 100 μg/mL. This starter culture was grown for 16 h at 250 RPM at 37 °C. All conditions were run in triplicate in 250 mL Erlenmeyer flasks, with 2.5 mL of a starter culture inoculating 47.5 mL media (types: LB media, fresh BMMY, spent BMMY, supplemented BMMY) containing 2% glycerol and 100 μg ampicillin. Flasks were grown at 37 °C with shaking at 250 RPM, with $OD_{600\ nm}$ measurements taken every 30 min. Once an $OD_{600\ nm}$ of 0.4 was reached, mCherry-EF2 expression was induced with the addition of Isopropyl β-d-1-thiogalactopyranoside (IPTG) at a final concentration of 1 mM. The cultures were then grown at 30 °C for 18 h. Cultures were spun at $4000\times g$ for 20 min at 4 °C to harvest the bacteria. Bacterial pellets were resuspended in a 10 mL lysis solution (10 mM Tris, 2 mM $CaCl_2$, pH 8). Cells were sonicated on ice, and the sonicate was added to 25 mL of boiling (>85 °C) lysis solution in a beaker. Once the temperature exceeded 85 °C again, one minute elapsed before removal from heat to boil out non-thermostable *E. coli* proteins [9]. The insoluble protein was spun down at $11,000\times g$ for 30 min at 4 °C. The supernatant was kept for analysis, where absorbance at 585 nm was read and recorded for mCherry-EF2 yield calculations. The final protein concentration was determined using the following equation: Absorbance at 585 nm/extinction coefficient of 44,854.2, multiplied by the molecular weight of mCherry-EF2 of 33,667.45. This value was related back to the volume of the post-boil solution, along with the final volume in the expression media flasks following the 18 h incubation, to give the total yield of the culture.

*2.6. Bioreactor Fermentation of YSM-fed E. coli*

The bioreactor experiment was carried out using a Bionet(R) Baby-F0 stirred tank reactor (Bionet, Murcia, Spain) with a working volume of 1.5 L. The system was autoclaved prior to use. This was followed by the addition of 500 mL of LB, YSM supplemented with 2 mM magnesium sulphate or YSM supplemented with 2 mM magnesium sulphate and 1% tryptone (*w/v*) to the bioreactor; the spent media were first filter-sterilised using 0.22-micron vacuum filters (Millipore, Cork, Ireland), and 2% glycerol and 100 μg/mL ampicillin were added. The system parameters used were as follows: temperature = 37 °C, pH = 7, dissolved oxygen (DO) = 20%, air = 0.5 slpm and stirring = 500 rpm. A cascade was set up for agitation (1500 rpm max) and air (1–10 slpm) for when DO drops below 20%, with agitation set up to be activated first. The acid used for pH balancing was 15% (*v/v*) sulphuric acid, and the base used was 1 M sodium hydroxide. The starter culture (50 mL with the same ingredients as used above of 2% glucose and 100 μg/mL ampicillin, grown overnight for 16 h at 37 °C and 250 rpm) was added to the bioreactor media (500 mL) once the temperature and pH were balanced within the instrument. This was considered as the time point 0. The culture was grown until the $OD_{600 nm}$ was > 10, with measurements taken every hour. A 1 mL sample was also taken every hour and centrifuged at 14,000× *g* for 3 min. The pellet of this sample was freeze-dried to measure cell dry weight, and 400 μL of the clarified supernatant of this sample was harvested for HPLC analysis of carbon source concentration [25]. Once the correct OD was reached, 1 mM IPTG was added to the culture with an additional 2% glycerol. From this point, an 18.5 h expression was carried out with the temperature dropped to 30 °C and left overnight. At the end of the expression, three 5 mL technical replicate samples were taken for protein purification and yield measurement by centrifuging the samples at 4000× *g* for 20 min and discarding the supernatant, and then the resulting biomass pellet was frozen at −80 °C overnight and stored at −20 °C until purification could be carried out [26,27].

**3. Results**

*3.1. Elemental and Amino Acid Composition Analysis of Fresh and Spent Media*

We first compared the elemental composition of LB media standardly used to grow *E. coli* to spent yeast media (YSM) harvested from shake flask cultures of a 48 h *K. phaffii* fermentation grown in BMMY (Table 1). Elemental composition analysis with ICP-MS identified significantly lower concentrations of sodium and magnesium and higher levels of phosphorus, sulphur, potassium and calcium in YSM compared to basal concentrations in fresh LB. The levels of magnesium and calcium measured in YSM were also lower than starting concentrations in fresh BMMY (Table S1), while YSM had similar levels of sodium, sulphur, phosphorus, potassium or calcium to BMMY. Some significant differences were also measured in levels of manganese, iron, zinc, aluminium, nickel and barium (Table S1). Magnesium and sodium were identified as the specific elements to replenish in subsequent supplementation experiments. The amino acid concentrations were all significantly lower in YSM when compared with LB, with the exception of cysteine and to a lesser extent glutamic acid, and a general supplementation approach was taken to restore amino acid levels using supplementation with 1% tryptone (*w/v*), normally used in the preparation of LB medium.

**Table 1.** Concentrations of elements (mg/L) measured using ICP-MS and amino acids (mM) measured using LC-MS/MS. Concentrations shown are basal for LB and recorded after 48 h *K. phaffii* culture for YSM.

| Element | LB (mg/L) | YSM (mg/L) |
|---|---|---|
| Sodium | 362.25 | 34.36 |
| Magnesium | 0.51 | 0.27 |
| Phosphorus | 18.07 | 408.62 |
| Sulfur | 8.77 | 311.66 |
| Potassium | 32.96 | 616.72 |
| Calcium | 0.49 | 1.38 |
| **Amino acid** | **LB (mM)** | **YSM (mM)** |
| Alanine | 2682.33 | 50.07 |
| Arginine | 523.33 | 4.91 |
| Asparagine | 607.00 | 35.23 |
| Aspartate | 1232.67 | 17.03 |
| Cysteine | 38.50 | 38.73 |
| Glutamate | 27.70 | 19.23 |
| Glutamine | 2848.00 | 84.10 |
| Glycine | 1258.33 | 17.97 |
| Histidine | 425.00 | 3.78 |
| Isoleucine | 1784.67 | 30.77 |
| Leucine | 4179.33 | 60.83 |
| Lysine | 2397.00 | 1482.33 |
| Methionine | 717.33 | 300.33 |
| Phenylalanine | 1518.33 | 765.33 |
| Proline | 569.00 | 22.97 |
| Serine | 1489.33 | 13.37 |
| Threonine | 1204.67 | 10.63 |
| Tryptophan | 526.00 | 3.13 |
| Tyrosine | 413.00 | 161.67 |
| Valine | 1250.67 | 35.80 |

*3.2. Supplementation in Shake Flask Experiments*

Growth rates were compared for eight culture conditions, three without supplementation and five with versions of YSM with supplementation (Figure 1). The specific growth rate measurements are tabulated in panel B. These data show that supplementation of yeast spent media (YSM) with magnesium, with magnesium and sodium or with magnesium sodium and amino acids significantly increases the growth rate of *E. coli* cultures over the growth rate of cultures grown in YSM alone and approaches the growth rate of cultures grown in LB.

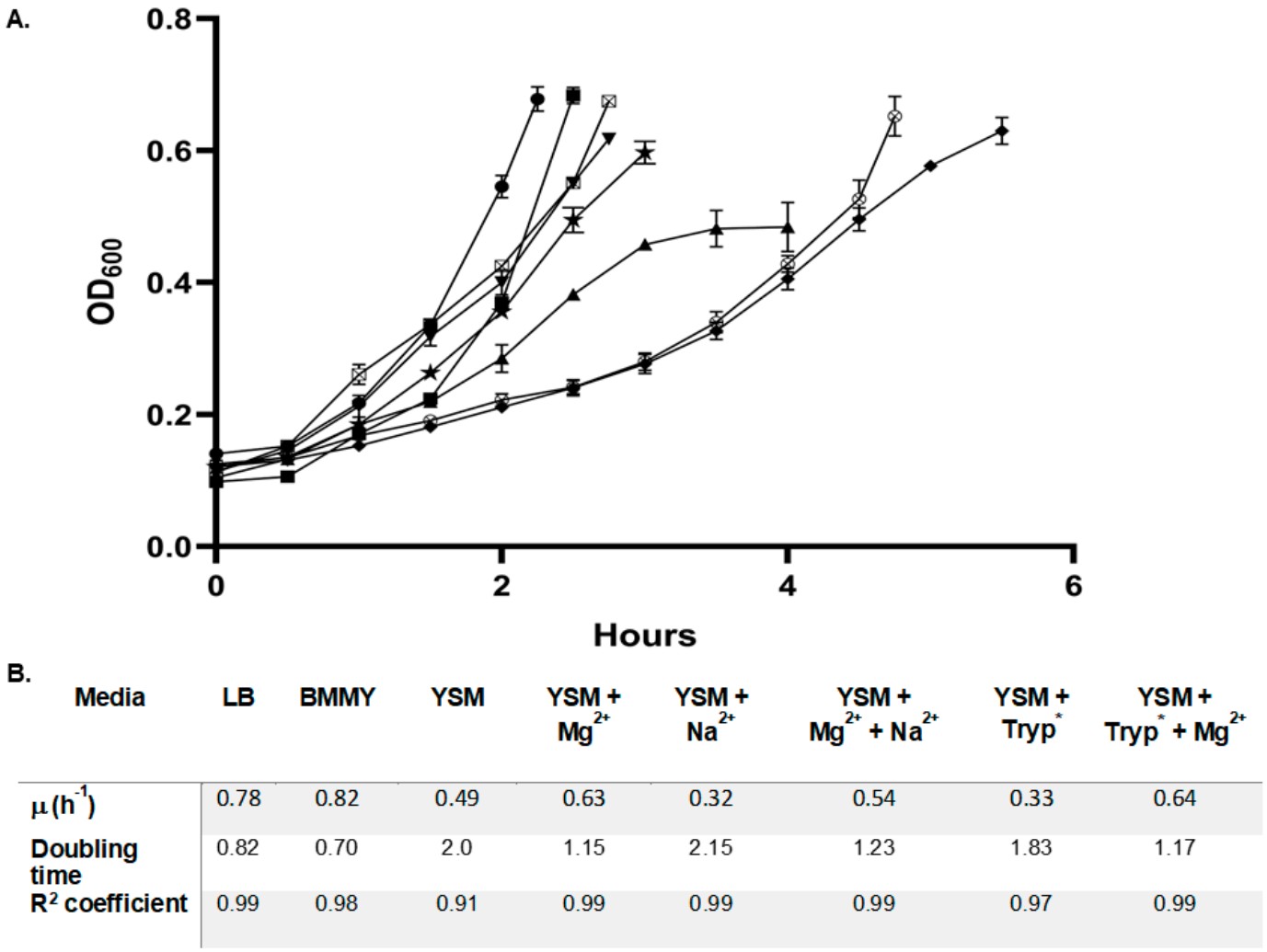

**A.**

**B.**

| Media | LB | BMMY | YSM | YSM + $Mg^{2+}$ | YSM + $Na^{2+}$ | YSM + $Mg^{2+} + Na^{2+}$ | YSM + Tryp* | YSM + Tryp* + $Mg^{2+}$ |
|---|---|---|---|---|---|---|---|---|
| $\mu (h^{-1})$ | 0.78 | 0.82 | 0.49 | 0.63 | 0.32 | 0.54 | 0.33 | 0.64 |
| Doubling time | 0.82 | 0.70 | 2.0 | 1.15 | 2.15 | 1.23 | 1.83 | 1.17 |
| $R^2$ coefficient | 0.99 | 0.98 | 0.91 | 0.99 | 0.99 | 0.99 | 0.97 | 0.99 |

**Figure 1.** Growth curves of *E. coli* BL21 DE3 in culture media. Cultures grown in triplicate, LB (solid circles), BMMY (solid squares), YSM (solid triangles), YSM + magnesium (inverted triangles), YSM + sodium (diamonds), YSM + magnesium + sodium (stars), YSM + tryptone (open circles), YSM + magnesium and tryptone (open squares). * Tryptone amino acid donor. Error bars are representative of the standard deviation of triplicate cultures for each condition.

*3.3. Determination of Recombinant Protein Expression Titres in Shake Flask Cultures*

Once each culture in the screen reached an $OD_{600 \ nm}$ of 0.6, expression of mCherry-EF2 was induced with the addition of IPTG and protein expressed for 18 h (except culture grown in YSM, which plateaued at $OD_{600 \ nm}$ of 0.5 and was subsequently induced early). Following this, the average titre of each condition was calculated from the purification supernatant. Titres of recombinant protein produced in each condition (Figure 2) highlight the poor nutritional value of YSM alone. Supplementation with 2 mM magnesium doubles the amount of protein expressed, while 14 mM sodium supplementation has little effect on the quantity of protein produced. This finding is reflected in the titre achieved in the YSM plus magnesium and sodium condition, which is equal to that of magnesium alone. Without magnesium supplementation, the YSM plus amino acids provided by tryptone performed very similarly to those with additional magnesium alone, with an average titre of approx. 0.182 g/L (versus 0.176 g/L and 0.183 g/L for the magnesium and sodium and magnesium conditions, respectively). The optimal YSM supplementation condition in the screen, yielding the highest titre of mCherry-EF2, was that of YSM with tryptone and magnesium supplementation combined. This condition showed no significant difference

from LB and was chosen alongside the magnesium-alone condition to be scaled up for bioreactor fermentation.

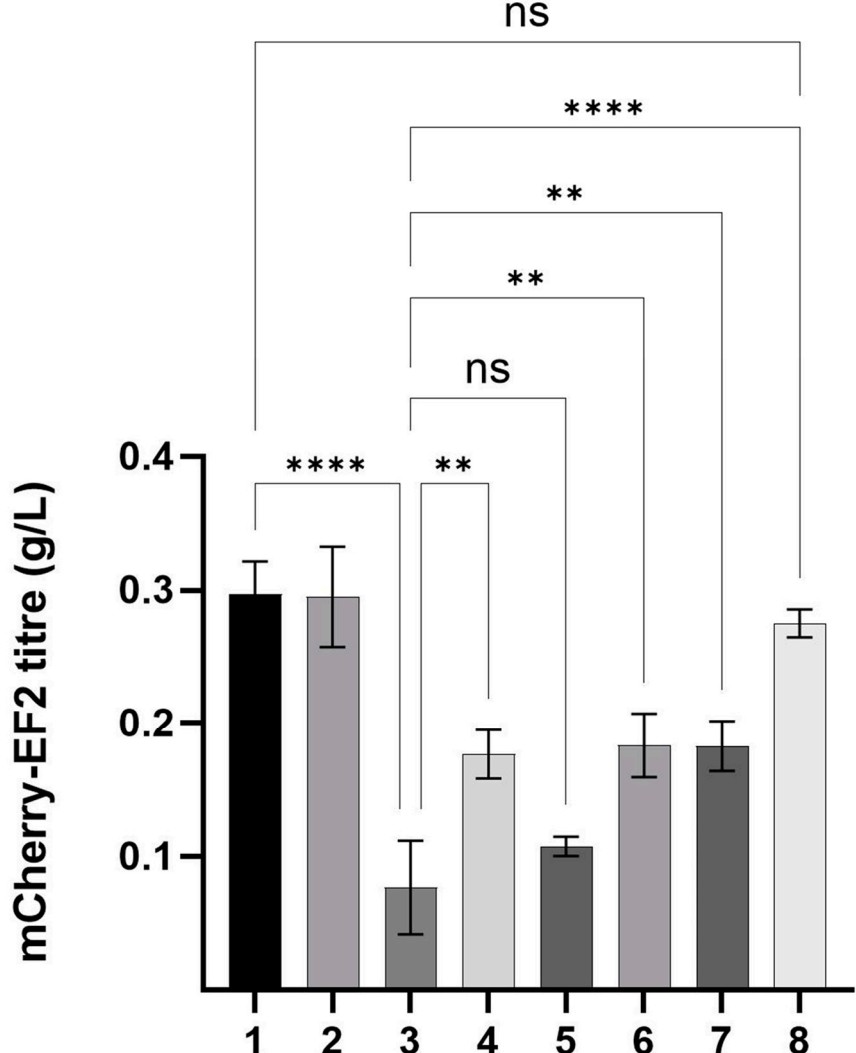

**Figure 2.** Titres of recombinant mCherry-EF2 protein following purification of triplicate cultures of each media condition. 1. LB, 2. BMMY, 3. YSM, 4. YSM + magnesium, 5. YSM + sodium, 6. YSM + magnesium and sodium, 7. YSM + tryptone, 8. YSM + magnesium and tryptone. Significance is determined using a two-way ANOVA with Tukey's post-analysis test, where the mean of each sample is compared. *p*-values are represented as follows: $p \geq 0.05$, ns (not significant) ** $p \leq 0.01$; **** $p \leq 0.0001$.

### 3.4. Recombinant Protein Expression Titres in Bioreactor Fermentation

*E. coli* grown in YSM supplemented with 2 mM magnesium and in YSM plus 2 mM magnesium and 1% (*w/v*) tryptone in shake flasks grew at a faster rate and expressed a higher titre of recombinant protein, and these conditions were next tested in scale-up experiments in a 0.5 L culture in a bioreactor. A target $OD_{600\,nm}$ of 10 was set as the induction point. However, the growth of the YSM plus magnesium-fed *E. coli* plateaued at an $OD_{600\,nm}$ of 6 (Figure 3A). This culture produced a titre of 0.79 ($\pm$0.04) g/L of mCherry-EF2 while cultures fed YSM plus 2 mM magnesium and 1% tryptone (*w/v*) produced a significantly increased titre of 2.29 ($\pm$0.02) g/L when compared with a titre of 1.29 ($\pm$0.09) g/L from cultures fed with LB (Figure 3B,C). The YSM plus magnesium-fed condition produced a biomass or cell dry weight (CDW) of 4.2 g/L, whereas supplementation of YSM with amino acids through the addition of 1% (*w/v*) of tryptone produced a 2.7-fold increase in CDW at 11.5 g/L, significantly higher (1.6-fold) than cultures grown in LB (7.1 g/L)

(Figure 3C). The YSM plus magnesium-fed culture produced a final mCherry-EF2 titre of 0.79 (±0.04) g/L (Figure 4B), which was 19% of the cell dry weight composition compared to 18% of CDW or 1.29 (±0.09) g/L for LB-fed fermentation (Figure 3C). Supplementation of amino acids with tryptone produced a titre of 20% of the CDW or 2.29 (±0.02) g/L, an increase of approx. 1 g/L over LB. LB and YSM supplemented with magnesium and amino acids shared equivalent productivity, measured in g/g/h (Figure 3C).

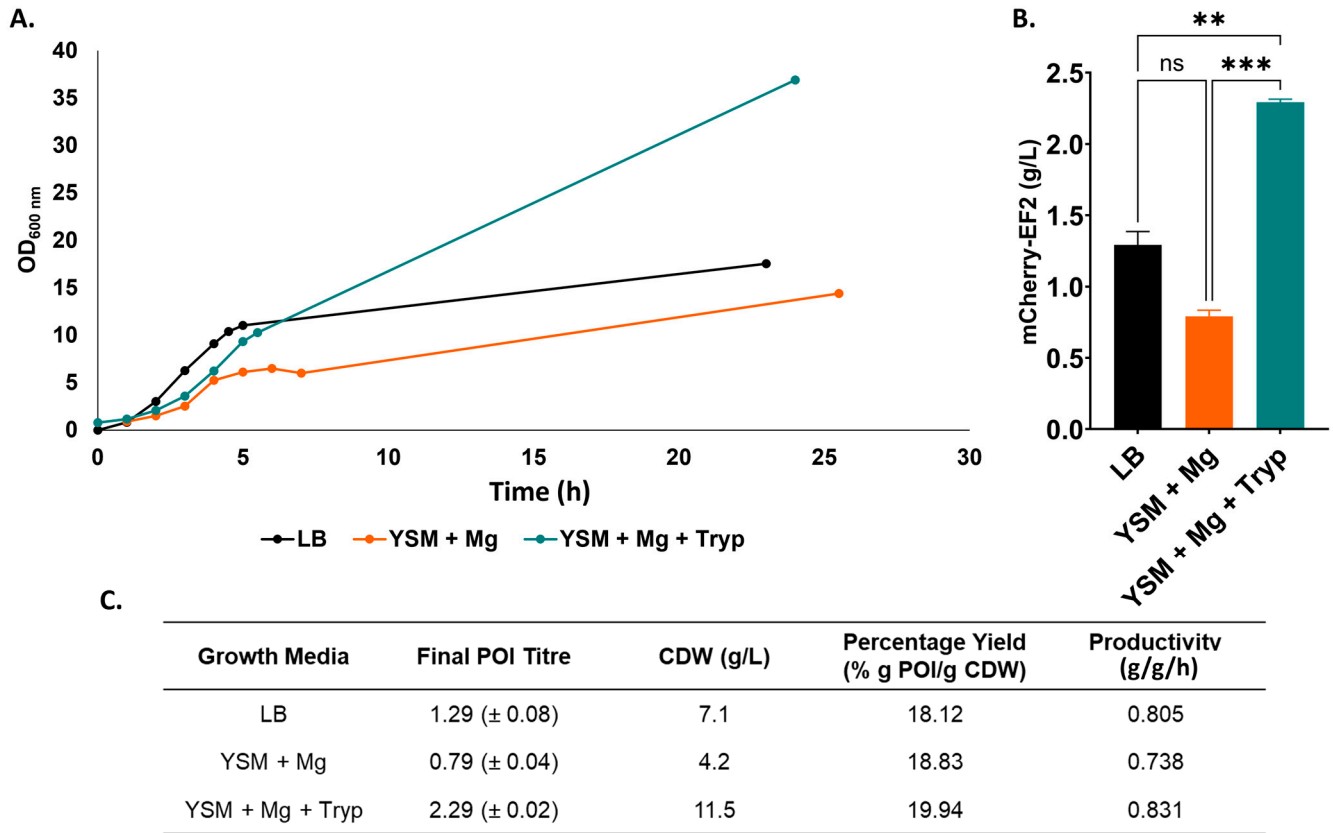

**Figure 3.** Recombinant protein expression in bioreactor fermentation with optimised supplementation. (**A**) Growth curves of *E. coli* cultures fed in supplemented media and in LB. (**B**) Titres of mCherry-EF2 expressed in each media condition, ns (not significant); ** $p \leq 0.01$; *** $p \leq 0.001$. (**C**) Calculations of percentage yield and productivity of each culture in terms of cell dry weight at the end of fermentation.

The YSM plus magnesium-fed culture consumed approximately 5 g/L of glycerol from the induction point to the end of the expression (18.5 h) (Figure 4A), while approximately double the amount (10 g/L) of glycerol was consumed in cultures fed with YSM plus magnesium and amino acids (Figure 4B). This higher consumption correlates directly with the higher CDW obtained in this culture (11.5 versus 4.2 g/L) (Figure 3C). A residual level of glucose from the starter culture was observed in the YSM plus magnesium condition. This was exhausted before the induction timepoint, at the same time as the growth plateau was seen in this condition (Figures 3A and 4A).

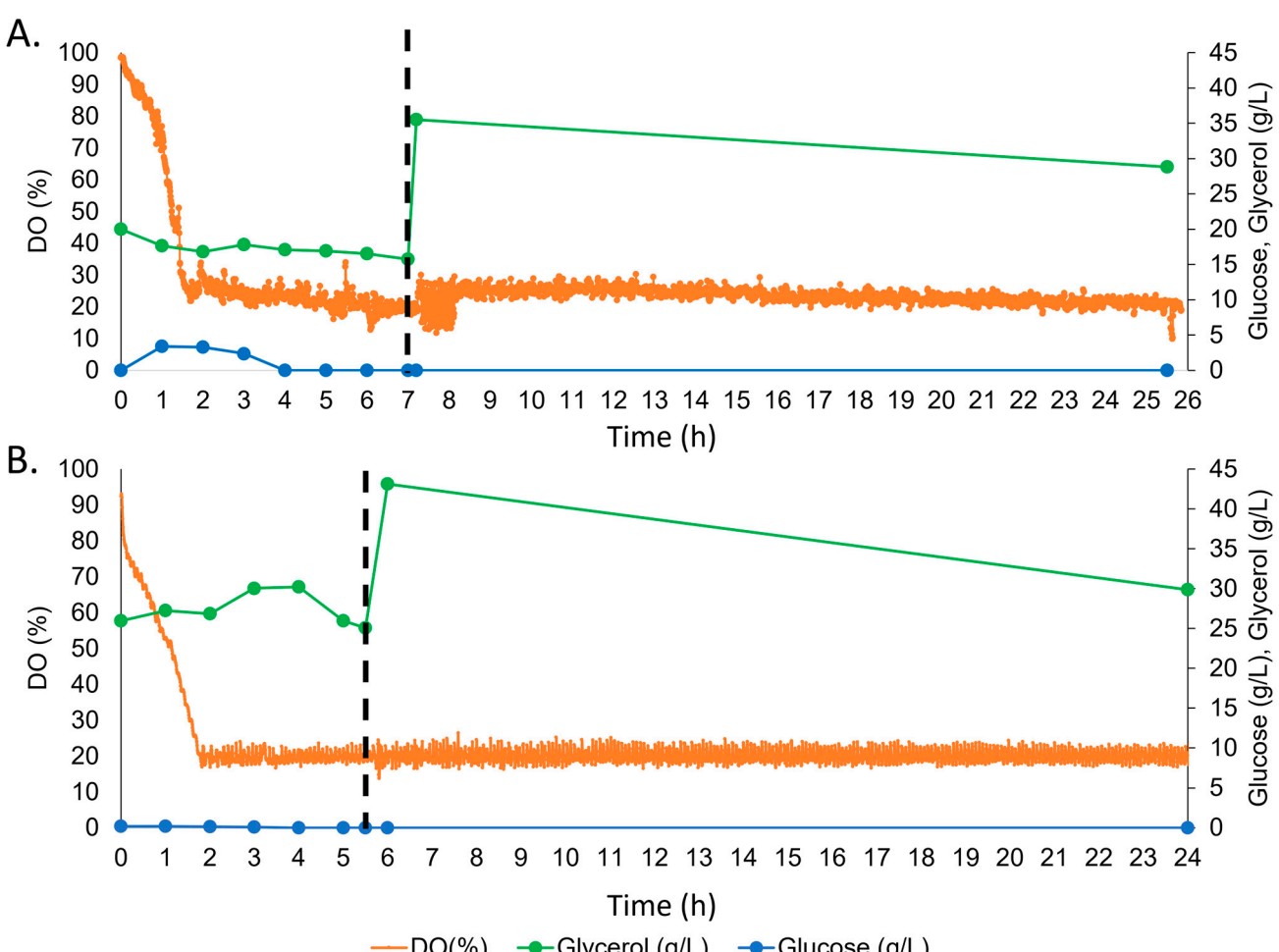

**Figure 4.** Bioreactor dissolved oxygen over time and HPLC analysis of carbon sources. (**A**) YSM + magnesium condition and (**B**) YSM + magnesium + tryptone. Bioreactor timelapse of dissolved oxygen (DO in percentage, orange line) and carbon source consumption measured by HPLC (glycerol and glucose, green and blue, respectively). IPTG induction of protein expression was carried out along with addition of 2% glycerol once cultures reached $OD_{600\,nm}$ of 10, marked with a dashed line, from which point expression was carried out over 18.5 further hours.

## 4. Discussion

The goal of this study was to evaluate the potential of valorising spent media from yeast fermentation (YSM) by studying the supplementation requirements to repurpose it to efficiently feed secondary fermentation of recombinant-protein-producing *E. coli*. Significant volumes of yeast culture media are consigned to waste treatment across the globe annually, meaning all remaining nutrients, water and energy used to create it are immediately lost after primary culture. The opportunity to reuse this resource in a meaningful way is a positive step towards implementing the aims of the circular bioeconomy to reduce global waste [28].

### 4.1. Spent Media Analysis of YSM Enabled Efficient Reuse of This Waste Resource

Comparative analysis of the elemental composition of spent yeast media with LB media, a standard feed for *E. coli* fermentation, identified changes in levels of elements that indicated where supplementation would most likely be beneficial. Levels of elements that notably increased in the yeast culture media following *K. phaffii* growth were phosphorus, sulphur and potassium with 6.8, 16.9 and 4.9% increases, respectively, over concentrations found in BMMY (Table S1). These increases are clearly a result of the growth of the yeast

culture conditioning the media and are most likely due to the lysis of *K. phaffii* cells releasing these elements into the media as there was no active secretion behaviour of the cells. *K. phaffii* have been described to have high levels of phosphorus and potassium (US Department of Agriculture: https://fdc.nal.usda.gov/fdc-app.html#/food-details/175043/nutrients, accessed on 30 May 2024). Concentrations of magnesium and sodium were both depleted in YSM when compared with starting concentrations of these elements in BMMY and LB (Tables S1 and 1). *E. coli* requires around 1.5 mg of magnesium per gram of cell dry weight produced [29]. The ICP-MS measurements on LB pre- and post-*E. coli* culture showed a consumption of over 98% of the starting concentration of magnesium by the *E. coli* used in this study (Table S2). In general, 2–5 mM of magnesium is accepted as the optimum for *E. coli* growth, and taken together with our measurements, this determined the supplementation with magnesium as the strategy carried out in this study [30–32]. Shake flask fermentation experiments confirmed that supplementation with magnesium improved two-fold the specific growth rate (m, generations per hour) and the recombinant protein yield over unsupplemented YSM, while sodium supplementation had a much lower effect (Figure 1), suggesting that magnesium supplementation is the most important elemental addition for successful *E. coli* growth in spent yeast media. An analysis of the amino acid content of YSM using LC-MS/MS showed the clear impact of primary yeast fermentation with all amino acids significantly depleted with the exception of cysteine. Metabolic analysis of LB media pre- and post-*E. coli* culture identified depletion of most amino acid levels with the exception of cysteine levels that increased and isoleucine levels that were reduced to a very small degree (Table S2). We anticipated that the concentrations of glutamate, glycine and lysine would have a significant impact on the rate of expression of mCherry-EF2 in the secondary bacterial fermentation as these are present in the primary sequence at levels of 10% each [33,34]. Levels of lysine remained at a high concentration in the YSM post-*K. phaffii* culture while glutamate and glycine were significantly depleted (Table 1). Aspartate, threonine, phenylalanine, serine and glutamate can all serve as precursors for the synthesis of other amino acids with only phenylalanine remaining at high levels in YSM. Serine is a precursor of glycine, while glutamate can be sourced from glutamine through the GOGAT pathway, though this is mostly glucose-dependent [35,36], and therefore, its synthesis may be impeded by glycerol as a primary carbon source in these cultures. The metabolic demand of having to synthesise most amino acids from de novo pathways likely contributed to the reduction in growth rate seen in the unsupplemented YSM-fed cultures in the shake flask culture experiments (Figure 1). The supplementation of amino acids either in defined amounts (via stoichiometric analysis [30,31]) or by addition of a complex additive rich in amino acids (tryptone) was likely to have a significant impact on increasing growth rate and productivity. It was interesting to observe that the addition of amino acids to YSM without magnesium supplementation was not sufficient to restore the growth rate or increase recombinant protein expression to a level higher than that for magnesium supplementation alone (Figures 1 and 2). It is clear from these experiments that both supplements are required to effectively valorise the media for *E. coli* fermentation. The product titre obtained with the addition of 2 mM magnesium alone provided a significant improvement over unsupplemented YSM, and with the addition of amino acids, a significantly increased titre of recombinant protein, equivalent to the rich microbiological media LB in shake flasks, was seen (Figure 2).

### 4.2. Titre of Recombinant Protein Is Increased with Optimised Supplementation of YSM in Bioreactor Fermentation

This supplementation with 1% tryptone (*w/v*) is an obvious way of restoring amino acid levels, is a standard ingredient in LB media, and would be expected to boost the growth and productivity of the culture. However, the impact of supplementation with magnesium and tryptone together on recombinant protein expression in the bioreactor was not expected with a high titre of 2.29 ($\pm$0.02) g/L, compared to a titre of 1.29 ($\pm$0.09) g/L from cultures fed with LB (Figure 3B). The percent recombinant protein in grams per gram

of cell dry weight obtained for YSM plus magnesium-fed culture (18.83%) was equivalent to LB (18.12%), albeit with a much lower CDW of 4.2 vs. 7.1 g/L [8]. The same measurement showed improved performance of the magnesium- and tryptone-supplemented YSM, with an increase of almost 2%, 19.94% per gram of CDW, over that of LB (Figure 3C). The most striking observation was the 2.7-fold increase in CDW of this culture over magnesium supplementation alone, confirming the critical need for amino acid supplementation but also identifying a very high performance of the supplemented waste media over LB with the increased CDW and productivity combining to deliver a final product titre of 2.29 g/L versus 1.29 g/L for LB-fed bioreactor fermentation. This straightforward supplementation strategy of the addition of 2 mM magnesium ($Mg_2SO_4$) and restoration of amino acid levels (1% tryptone (*w/v*)) provides a very clear and inexpensive route towards the valorisation of spent yeast media from industrial processes. The restoration of amino acid levels with protein extracts from other waste streams, e.g., cheese manufacture by the dairy industry, may be explored to enhance the benefits for the circular bioeconomy.

An interesting observation of the culture fed with YSM plus magnesium alone was while 38% of total protein was made up of the recombinant product, a clear inhibition of the culture growth rate and almost complete cessation of growth in the bioreactor was observed. It can be seen from the HPLC measurements (Figure 4A) that the culture was still increasing its respiration at this point as the dissolved oxygen (DO) continued to drop despite the slowed growth rate (Figure 4). This respiratory shift may therefore be a response to some other energy consumption needs.

One explanation for this sudden respiratory shift is the auto-induction of the expression of the recombinant protein, mCherry-EF2. Glycerol was the primary carbon source used in the expression of the secondary culture chosen over glucose as it aids the maintenance of high cell density, reduces acetate accumulation, reduces sequestration of recombinant protein products into inclusion bodies and is a low-cost supplement [32–35]. This also removes the inhibition of the T7 expression system by catabolite repression [30]. Some residual glucose remains at the beginning of the culture, leftover from the starter culture addition directly into the bioreactor. However, this was fully metabolised before the induction point and may have caused the cultures to initiate protein expression earlier. With the limiting amounts of amino acids available in the YSM, this could have resulted in *E. coli* synthesising these protein building blocks through de novo pathways. These alternate biosynthesis pathways are known to have a higher energy demand than simply metabolising the necessary amino acids from the media [37–39]. This further helps to explain the boost in growth rate and increase in protein titre seen when the YSM was also supplemented with amino acids from tryptone. Cultures grown in YSM + 2 mM magnesium and 1% tryptone (*w/v*) growth media delivered a final protein titre of 2.29 g/L, approximately 1.8-fold higher than that of 1.29 g/L from LB-fed cultures. In terms of total protein, 40% of the total protein was mCherry-EF2, only slightly higher than the yield obtained by YSM + 2 mM magnesium (38%) and LB (36%). This result indicates that the 1.8-fold increase in final yield seen in the YSM + 2 mM magnesium and 1% tryptone (*w/v*)-fed culture was mainly due to the significantly higher biomass obtained in this condition.

## 5. Conclusions

We demonstrate that yeast spent media are viable media for reuse in feeding secondary *E. coli* cultures with minimal supplementation steps. These cultures can be highly productive, with the correct balance of supplementation driving more productive secondary fermentation. In terms of wider implications, the WARIEN and PMIs of various bioprocesses could be substantially improved as well as contribute to a reduction in cost by incorporating the reuse of spent media. To apply the findings of this work, the preparation of YSM for use in *E. coli* fermentations is relatively straightforward and is compatible with industrial-scale bioprocesses. Clarification of spent media of biomass can be performed with depth filtration. Further filtration steps may also be added that would necessitate further analysis of the filtrate for changes in elemental and amino acid composition. Sup-

plementation is subsequently performed with the addition of sterile formulations of liquid MgSO$_4$, NaCl and tryptone solutions. This study therefore represents another step towards a more circular bioeconomy for the future, with the utilisation of industrial waste streams and cost-reduced bioprocesses.

**Supplementary Materials:** The following supporting information can be downloaded at https://www.mdpi.com/article/10.3390/applmicrobiol4020065/s1: Table S1: Concentrations of elements in fresh BMMY and YSM post-yeast culture using ICP-MS. Table S2: Concentrations of amino acids and elements in LB pre- and post-*E.coli* culture measured by LC-MS/MS and ICP-MS/MS.

**Author Contributions:** D.J.O. designed the study. C.D.L. and L.M. performed the experiments. C.D.L., L.M. and D.J.O. analysed the data. C.D.L., L.M. and D.J.O. wrote the paper. All authors have read and agreed to the published version of the manuscript.

**Funding:** This research was funded by the Atoms-2-Products centre for doctoral training, which is supported by the Science Foundation Ireland (SFI) and the Engineering and Physical Sciences Research Council (EPSRC) under Grant No. 18/EPSRC-CDT/3582.

**Data Availability Statement:** All data are provided in the manuscript and Supplementary Materials.

**Acknowledgments:** Our thanks to Katalin Kovacs, Peter License and the ICP-MS suite at the University of Nottingham for their assistance in the analysis of our media composition by ICP-MS. Our thanks to Lorraine Brennan and Xiaofei Yin at the Core Facilities in the Conway Institute for their assistance in the metabolomic analysis of the media supernatant.

**Conflicts of Interest:** The authors declare no conflicts of interest.

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
