# Peer review of "Valorisation of Spent Yeast Fermentation Media through Compositional-Analysis-Directed Supplementation"

_2673-8007, doi:10.3390/applmicrobiol4020065_

Round 1
Reviewer 1 Report (Previous Reviewer 1)
Comments and Suggestions for Authors
The new version of the manuscript "Valorization of spent yeast fermentation media through compositional analysis-directed supplementation" has much improved compared to the previous one. The authors analyze elemental and amino acid variation in Yeast spent media and obtain a new effective medium for E. coli recombinant protein production.
- The authors should discuss how it is possible to obtain an increase in some elements after Komagatella phaffi growth.
- They should also discuss the practical application of YSM in E. coli fermentations. What additional processes need to be carried out (filtration, etc.) and how might this affect the economic viability of the process?
- IMPORTANT: The improvement in growth and protein production is only based on the 25h time point (Figure 3). In the earlier time points, no improvement can be observed due to YSM supplementation. Therefore, the authors should strengthen this conclusion by providing additional data (I think they have them as Fig. 3b has SD). How many experiments were carried out? Please provide statistics for this time point.
Author Response
- The authors should discuss how it is possible to obtain an increase in some elements after Komagatella phaffi growth.
Levels of elements that notably increased in the yeast culture media following K. phaffii growth were phosphorus, sulphur and potassium with a 6.8, 16.9 and 4.9 % increase respectively over concentrations found in BMMY (Table S1). These increases are clearly a result of the growth of the yeast culture conditioning the media and are most likely due to lysis of K. phaffii cells releasing these elements into the media as there was no active secretion behaviour of the cells. K. phaffii have been described to have high levels of phosphorus and potassium (US Department of Agriculture: https://fdc.nal.usda.gov/fdc-app.html#/food-details/175043/nutrients). Concentrations of magnesium and sodium were both depleted in YSM when compared with starting concentrations of these elements in BMMY and LB (Table S1 & table 1).
This text has been added to the Discussion section between lines 275-285 in the revised manuscript
- They should also discuss the practical application of YSM in E. coli fermentations. What additional processes need to be carried out (filtration, etc.) and how might this affect the economic viability of the process?
To apply the findings of this work, the preparation of YSM for use in E. coli fermentations is relatively straightforward and is compatible with industrial scale bioprocesses. Clarification of spent media of biomass can be performed with depth filtration. Further filtration steps may also be added that would necessitate further analysis of the filtrate for changes in elemental and amino acid composition. Supplementation is subsequently performed with addition of sterile formulations of liquid MgSO4, NaCl & tryptone solutions.
This text has been added to the Conclusions between lines 383-389.
- IMPORTANT: The improvement in growth and protein production is only based on the 25h time point (Figure 3). In the earlier time points, no improvement can be observed due to YSM supplementation. Therefore, the authors should strengthen this conclusion by providing additional data (I think they have them as Fig. 3b has SD). How many experiments were carried out? Please provide statistics for this time point.
The improvement in growth rate was evident from 5 hours into the culture where the growth rates for cultures grown in bioreactors fed with LB and YSM + Magnesium and Tryptone have reached an OD600 of 10 and are induced with IPTG. YSM + Magnesium alone is slower growing and was induced 2 hours later without reaching this OD. Both LB and YSM + Magnesium alone fed cultures respectively begin to plateau whereas the growth of YSM + Magnesium and Tryptone fed culture continues to increase to the 25 hour timepoint. These are single bioreactor experiments. The standard deviation shown in the protein yields (Fig 3B) are from three technical replicates for each condition, where the protein purification was performed in three separate experiments with samples taken from each bioreactor condition.
Reviewer 2 Report (Previous Reviewer 2)
Comments and Suggestions for Authors
All my concerns have been resolved, I think this manuscript can be accept.
Comments on the Quality of English Languageno
Author Response
Thank you very much for your positive review
This manuscript is a resubmission of an earlier submission. The following is a list of the peer review reports and author responses from that submission.
Round 1
Reviewer 1 Report
Comments and Suggestions for Authors
The authors of the manuscript "Compositional analysis of spent media from yeast fermentation identifies supplementation strategies supporting secondary fermentation. " consider the interesting possibility of using spent media from Pichia pastoris fermentation for recombinant protein production in E. coli in a secondary fermentation. The basic idea is in principle appealing, but it turns out to be basically unfeasible as many nutrients, especially amino acids, are consumed during the primary fermentation. As the authors hypothesize, this leads to an increased metabolic load for E. coli, which cannot reach the production levels obtained with LB medium, even if the authors manage to improve the production by Mg++ supplementation.
MAIN POINTS
- The main problem with the manuscript is that the authors base their media supplementation strategy only on minerals. During a fermentation many other nutrients are depleted and a more comprehensive metabolomic approach would have been much more appropriate to design suitable supplements. An analysis of the major nutrients consumed during LB use would also have been much more useful.
- Terminology is used in a way that is not appropriate, or at least not common, in the biotechnology community. What the authors call "yield" (e.g., g/L) is more commonly referred to as "titre" or "production". Yield, on the other hand, is most often used to refer to a ratio; for example, biomass yield = g biomass/g substrate. The author refers to this as "productivity", which instead is expressed as g/l x h.
- The experiments reported in Figure 2, especially panel A, should be repeated. The starting OD is very different and is 0.4 for 2 of the conditions, which is too high considering that the induction is carried out at 0.6. This is particularly unacceptable since in this case the parameters are calculated based on 2 time points (if we think that it is reasonable to consider a lag phase in these cultures). The specific growth rate is reported as k (defined only in the Discussion). The specific growth rate is not the one reported by the authors. It is reported as h-1 and is often represented as mu, especially for microorganisms.
- Determination of batch effects. Many more batches should be tested to claim significant differences between them and to define the variability. Again, everything is based on minerals; what about amino acids?
- Line 393. Defining amino acid supplementation as "unnecessary" is clearly an overstatement. The authors should demonstrate this experimentally.
MINOR POINTS
- The introduction reports an extensive discussion about the circular economy of media production for biopharmaceutical production. I think readers would benefit from a more detailed economic perspective, perhaps shortening the text presented.
- Figure 1. Since the scale is logarithmic, some significant differences are hard to appreciate. I would suggest making a small bar chart for each element using different scales to better visualize the differences.
- Line 244: "While some levels of certain amino acids...". This is a very general sentence. Please be more specific.
- Figure 4. Panels A and B are not explained.
- Figure 5 legend; please correct "eyelid".
- Line 290. The authors refer to Figure 6 and state that the "YSM post-primary culture was high in phosphate...". However, Figure 6 shows phosphate in the precipitate.
Figure 7. Again, the log scale makes it difficult to assess the change in elemental content during bacterial cultivation.
- L.340 "... simply metabolizing...". I think "transport" is more appropriate.
Reviewer 2 Report
Comments and Suggestions for Authors
This study focused on evaluating the potential of supplementing a yeast spent media (YSM) for reuse in the production of a recombinant protein in bacteria. However, there are sever questions that might be the reviewer’ attention.
1. The tittle was too long.
2. There were too many keywords.
3. The genus name should be italic throughout the manuscript.
4. Fig 2A, why the time was different.
5. The discussion section was more like results.
6. It is unclear why E. coli was chosen for use in secondary fermentations.
7. Fig 5, Why there was no data at the latter half of the time? Fig 5C, lack of data analysis in the figure caption
8. Fig 6, no error bars?
9. The Y-axis of some figures should be of uniform size. Broken or right Y-axis can be used.
Comments on the Quality of English Languageno